# Weight Gain and Change in Body Mass Index after Age 20 in the Brazilian Population and Associated Sociodemographic Factors: Data from the National Health Survey

**DOI:** 10.3390/ijerph19052851

**Published:** 2022-03-01

**Authors:** Nathalia A. B. Souza, Karina A. Rimes-Dias, Janaina C. Costa, Daniela S. Canella

**Affiliations:** 1Postgraduate Program in Food, Nutrition and Health, Rio de Janeiro State University, Rio de Janeiro 20550-013, Brazil; brigidonathalia@gmail.com (N.A.B.S.); karinarimes@hotmail.com (K.A.R.-D.); 2International Center for Equity in Health, Federal University of Pelotas, Pelotas 96020-220, Brazil; calu.janaina@gmail.com; 3Department of Applied Nutrition, Institute of Nutrition, Rio de Janeiro State University, Rio de Janeiro 20550-013, Brazil

**Keywords:** obesity, excess weight, weight gain, health surveys, descriptive epidemiology

## Abstract

Obesity is considered one of the main contemporary public health problems. We aim to assess changes in body weight and nutritional status in adulthood and the associated sociodemographic variables. We use data from the 2013 National Health Survey (*n* = 21,743). Changes in weight and body mass index (BMI) were calculated based on mean difference between measurements at age 20 and data collected at the interview, stratified by sex. The association was analyzed using linear regression. Mean weight gain was greater among women than men. The largest gain was verified among the younger adults for both sexes. Age was found to be associated with weight and BMI change in men and women where, for every additional year of age, there was an increase in weight and BMI of 0.10 kg and 0.04 kg/m^2^ in men and of 0.22 kg and 0.09 kg/m^2^ in women, respectively. For education, a direct association was found for men and an inverse for women. Association with area of residence was significant among males only, where rural men gained less than their urban counterparts. Weight gain was progressive, being more marked in the younger group, and was associated with education differently according to sex.

## 1. Introduction

Obesity is considered one of the main contemporary public health problems due to its high prevalence and the impact it produces both at individual and collective levels. Obesity is a multifactorial disease and one of its main causes is inadequate diet [1]. Between 1975 and 2016, the prevalence of obesity almost tripled worldwide, contributing to an increase in global deaths and to the emergence of non-communicable diseases (NCDs) attributable to excess adipose tissue. In 2017, excess weight (overweight and obesity) was estimated to have caused 2.4 million deaths and 70.7 million debilitating diseases in females, and 2.3 million deaths and 77.0 debilitating diseases in men [2,3].

This rise in the prevalence of obesity can also be seen in Brazil. According to data from the Surveillance System of Risk and Protective Factors for Chronic Diseases by Telephone Survey (Sistema de Vigilância de Fatores de Risco e Proteção para Doenças Crônicas por Inquérito Telefônico-Vigitel), in 2006, a total of 11.4% of adults were obese [4]. Between 2010 and 2014, this rate rose from 15% to 18% in both sexes. In 2016, an estimated 18.9% of Brazilian adults had obesity, with slightly higher rates observed among women (19.6%) than men (18.1%) [5]. In 2019, there was a trend of rising obesity rates, where the condition affected 20.3% of individuals, with similar rates in men and women [6].

Despite the fact that in the last survey the prevalence of obesity was similar between men and women, this was not the trend in recent decades. Theories proposed for this difference point to the effect of specific influences on weight gain of women, such as pregnancy and menopause [7]. Weight retention during pregnancy may also represent a determinant of obesity in women [8]. Furthermore, one of the major deleterious effects of declining estrogen is a change in women’s body composition during menopause, altering the pattern of distribution of subcutaneous fat [9]. Besides biological factors, certain social aspects may also play a role in the greater weight gain among females. One hypothesis for this phenomenon is the difference in profession and family life dynamics for men and women. The pattern of inequality in the division of domestic tasks persists, even in developed countries, particularly when there are children, a spouse or an individual who needs special care. Thus, given the need to reconcile different demands, women may have less time for self-care, where this appears to have an adverse effect on health and on the capacity to maintain healthy life habits [10]. These aspects indicate the importance of investigating obesity by stratifying the analysis for sex.

A number of factors may have contributed to the occurrence of obesity, including poor diet, characterized by a switch from fresh or minimally processed food to ultra-processed foods [11,12,13]. Dietary patterns with a high intake of ultra-processed foods are typically rich in energy, free sugars, trans-fat and salt, and lacking in fiber and potassium. These characteristics can promote nutritional imbalances and impair the body’s ability to control energy balance, increasing the risk of excess weight gain [14,15]. In addition to diet and sociodemographic factors, genetic factors can also have a role, even on a smaller scale, in the onset of obesity [16].

In this respect, excess weight gain in adulthood can result in increased nutritional risk, where individuals hitherto classified as normal weight start to become overweight or obese. Although in Brazil the trajectory of overweight and obesity is well documented, there is a dearth of reports on population-based studies assessing weight gain and nutritional status changes throughout the life span and exploring sex differences. Based on a representative sample of the Brazilian population, the objective of the present study was to assess changes in weight and nutritional status in adulthood and to identify the sociodemographic factors associated with this shift, examining their influence on men and women.

## 2. Materials and Methods

A cross-sectional study of data from the 2013 National Health Survey (Pesquisa Nacional de Saúde—PNS) was carried out. The PNS is a household survey conducted by the Brazilian Institute of Geography and Statistics (IBGE) in collaboration with the Oswaldo Cruz Foundation (FIOCRUZ) and the Ministry of Health [17].

The sampling plan employed by the PNS was a three-stage cluster sampling with geographic and socioeconomic stratification of primary sample units. Census sectors were the primary sampling units, households the second stage, and dwellers aged >18 years the third stage. Overall, a total of 81,254 households were visited, 69,994 of which were occupied. A total of 64,348 household interviews and 60,202 individual interviews with dwellers were carried out [17].

The main questions of interest in this study pertained to data on weight at 20 years of age and at the time of interview (present), the height of the individuals, and some sociodemographic characteristics. The data on weight at 20 years was collected based on the questions: “Do you remember what your approximate weight was at around 20 years of age?”, “If yes, what was it?” and was applied only to respondents aged 30 or older. At the time of the interview, respondent weight and height were measured by trained researchers using a portable set of digital scales and a stadiometer.

Elderly people (>60 years old), pregnant women and individuals who did not answer the questions of interest were excluded from the analyses. Thus, the final sample comprised 21,743 adults, aged 30–59 years.

Estimates of body mass index (BMI) at 20 years and BMI at time of interview were calculated by dividing body weight (kg) by measured height squared (meters). Individuals were classified into: normal weight (BMI < 25 kg/m^2^), overweight (BMI ≥ 25 and <30 kg/m^2^) and obese (BMI ≥ 30 kg/m^2^), at the two assessment time points, as per criteria defined by the WHO [18].

For the estimate of differences in weight and BMI, differences between current weight and weight at age 20 and between current BMI and BMI at age 20 were calculated. Shifts in the nutritional status classification of the participants between the age of 20 and current age were analyzed and expressed as a percentage of individuals belonging to each BMI category.

The relationship of sociodemographic characteristics with weight gain and BMI increase was explored. The sociodemographic variables evaluated were: race/skin color (white, black, mixed-race, Asian and indigenous), age (stratified into: 30–34, 35–39, 40–44, 45–50, 51–54 and 55–59 years for the descriptive analyses, and used as a continuous variable in regression models), educational level (stratified into: 0–8, 9–11 and ≥12 years of formal study), and area of residence (urban or rural). According to IBGE, urban areas correspond to cities (municipal seats), towns (district seats) or isolated urban areas, while rural areas covered the entire area located outside these limits.

Descriptive statistical analysis of all variables was performed to determine the percentage distribution of the characteristics of interest in the study population. Categorical variables were expressed as relative frequency and continuous variables as mean and respective 95% confidence intervals (95% CI). All estimates of change in weight and BMI were calculated for Brazil and for each variable studied, stratified by sex. Significant differences were identified based on the comparison among the 95% CI. The absence of overlapping between intervals was assumed as a significant difference, considering the level of significance of 5%. Lastly, crude and multiple linear regression models were employed to assess the association of sociodemographic variables with changes in weight and BMI. Variables that were significant on crude models (95% CI did not include the null value) or that changed the adjusted coefficients were included in the multiple models. Adjustments were tested for each sex.

All data analyses were carried out using the statistics package Stata SE version 16.0 (Stata Corp., College Station, TX, USA), with the survey module considering the effects of complex sampling of the PNS, enabling extrapolation of the results for the whole adult Brazilian population.

## 3. Results

Of the 21,743 adults aged 30–59 years assessed, 55.1% were women and 44.8% were men. The age distribution declined with increasing age. The sample was predominantly white, followed by black and mixed-race, while Asian and indigenous categories accounted for 1%. The majority of individuals had 12 or more years of education. There were more dwellers from urban than rural areas (Table 1).

Weight and BMI distribution, at age 20 and at current age, according to sociodemographic characteristics and sex, are shown in Table 1. At 20 years, mean weight and BMI of men were 66.1 kg (95% CI 65.7; 66.5) and 22.4 kg/m^2^ (95% CI 22.3; 22.5), respectively, and at current age were 79.8 kg (95% CI 79.2; 80.3) and 27.0 kg/m^2^ (95% CI 26.9; 27.2), representing a mean BMI increase of 20.7%. For women, mean weight and BMI at 20 were 53.9 kg (95% CI 53.6; 54.2) and 21.3 kg/m^2^ (95% CI 21.2; 21.4), and at current age were 69.6 kg (95% CI 69.2; 70.1) and 27.5 kg/m^2^ (95% CI 27.3; 27.7), representing a 29.1% increase in mean BMI.

A negative relation was observed between mean weight at age 20 and age at the time of interview in both sexes: individuals younger at the time of interview reported greater weight at 20 compared to those older at the time of interview. This same age effect was seen for current BMI in both sexes. With regard to race/skin color, there were differences for weight at 20 and current weight between white and black groups, although the difference for BMI was only observed among men, with higher estimates in the white population. For education, men with more years of education had a greater mean weight than those with fewer years of education, both at 20 years of age and current age, with the same pattern found for current BMI. By contrast, women with more years of education had lower current mean weight and BMI at 20 and current BMI compared to those with lower educational level. Additionally, mean weight at 20 years, current weight and current BMI were higher among urban males compared to their rural counterparts (Table 1).

The results for estimated weight gain and difference in BMI in adulthood are given in Table 2. Men gained an average of 13.7 kg (95% CI 3.1; 14.2), or 4.6 kg/m^2^ (95% CI 4.4; 4.7). The estimated weight gain and BMI change for the women were 15.6 kg (95% CI 15.2; 16.0) and 6.2 kg/m^2^ (95% CI 6.0; 6.3), respectively.

A significant difference in the increase in weight and BMI was noted only among men aged 40–44 relative to those aged 30–34, and across all older relative to younger age brackets for women. From age 20 to 30–34 years, there was a weight gain of 12.1 kg for both sexes. Mean weight gain from age 20 to 55–59 years was 14.6 kg in men and 18.3 kg in women. Adults younger at the time of interview (30–34 years) were heavier at 20 years of age than individuals older at interview (55–59 years). This difference among males averaged 3.4 kg versus 3.6 kg among females (Table 2).

Males with ≥12 years of education had a higher mean difference in weight (14.8 kg; 95% CI 14.1; 15.5) relative to those with 0–8 years of education. In women, the largest difference in weight was found among those with 9–11 years of education (16.8 kg; 95% CI 15.7; 17.8) relative to women with ≥12 years of education. The difference in BMI showed similar results in both sexes. For the area of residence, weight and BMI differences were greater among men from urban areas, with differences in the weight of 14.2 kg (95% CI 13.6; 14.7), versus averages of 10.3 kg (95% CI 9.4;11.1). The BMI difference presented results in the same direction as the weight difference (Table 2).

Changes in the nutritional status of males from the age of 20 years to the current age are shown in Table 3. Of the men who had normal weight at 20 years, 45.4% (95% CI 43.3; 47.5) were overweight and 16.9% (95% CI 15.5; 18.4) had obesity in adulthood. Of the overweight individuals at 20 years, only 17.3% (95% CI 14.3; 20.7) lost weight to attain a BMI classified as normal, and 44.3% (95% CI 40.2; 48.6) went on to obesity. Of those who had obesity at 20 years, only 10.1% (95% CI 5.3; 18.2) attained normal weight, while 63.3% (95% CI 50.2; 74.6) remained obese. Thus, less than 40% of individuals of normal weight at 20 years maintained adequate nutritional status, whereas over 60% of individuals who had obesity as younger adults remained so in later life.

Comparing individuals aged 30–34 years and 55–59 years, in the older group a lower proportion of those of normal weight at 20 remained so in adulthood (30.9%; 95% CI 26.1; 36.0), while a greater proportion developed obesity later in adulthood (21.1%; 95% CI 16.6; 26.3). For education, a lower proportion of high-educated (≥12 years) individuals retained normal weight into adulthood (34.1%; 95% CI 31.5; 36.8), compared with lower educated (≤8 and 9–11 years) groups. No significant differences according to race/skin color were found. Results for the area of residence revealed that a higher rate of rural dwellers maintained normal weight status compared with urban dwellers (Table 3).

The data on changes in nutritional status among women are presented in Table 4. Of the normal-weight women at 20 years, 38.5% (95% CI 37.0; 40.1) remained of normal-weight, 37.1% (95% CI 35.7; 38.6) were overweight, and 24.2% (95% CI 22.8; 25.6) had obesity. Of those who had obesity at 20 years, only 10.9% (95% CI 6.8; 16.9) attained normal weight, while 61.4% (95% CI 52.5; 69.7) remained obese.

Similarly to the men, comparing women aged 30–34 and 55–59 years, in the older group a lower proportion of those of normal weight at 20 remained so in adulthood (28.5%; 95% CI 24.6; 32.6), while a greater proportion developed obesity (21.1%; 95% CI 16.6; 26.3). Conversely for education, a greater proportion of the high-educated women remained of normal weight into adulthood (43.6%; 95% CI 43.6; 45.8) and a lower proportion developed obesity (20.8%; 95% CI 19.1; 22.7), compared with low-educated groups (43.6%; 95% CI 41.5; 45.8 and 29.1%; 95% CI 26.5; 31.9, respectively). No significant differences for race/skin color or area of residence were found (Table 4).

The results for association of sociodemographic variables with change in weight and BMI by sex are given in Table 5. On the crude analyses, an association with age, education and area of residence was found in men and an association with age and education in women. After adjusting for significant variables on crude analysis, the same associations persisted, albeit weaker for some aspects.

On the adjusted models, age was associated with a change in weight and BMI in both sexes, exhibiting an increase of 0.10 kg (95% CI 0.04; 0.16) and 0.04 kg/m^2^ (95% CI 0.02; 0.06) for every additional year of age in men, versus 0.22 kg (95% CI 0.17; 0.26) and 0.09 kg/m^2^ (95% CI 0.07; 0.11) in women. Educational level was associated with an increase in weight, but not BMI, among men, where individuals with education ≥12 years exhibited a significant increase compared to those with 0–8 years of education. High-educated women (≥12 years of education) exhibited a decrease in BMI relative to those with lower education (0–8 years). Area of residence (rural or urban) was significantly associated with a change in weight and BMI among men only, where rural-dwelling men gained less weight and BMI compared to urban men (Table 5).

## 4. Discussion

Based on an analysis of representative data for the Brazilian adult population, less than 40% of individuals of normal weight at age 20 remained so after the age of 30, while over 60% of adults who had obesity at age 20 remained so into later life. These data highlight that the process of weight gain continued during adulthood. The data also showed different patterns of weight gain and changes in nutritional status between men and women according to socioeconomic characteristics.

Mean weight gain was greater among women than men, differing by approximately 2 kg. This result of greater weight gain among women is consistent with findings of other studies [5,19,20,21]. In this respect, the relationship between sex and weight gain is complex, involving social and biological issues, where no sole cause can be identified or degree of impact of these aspects established. However, stratified analyses exploring differences between men and women can promote investigations that are more in-depth or involve other fields of knowledge, in a bid to further understand the role of gender in this outcome.

Data from the Vigitel 2019 for the 26 major Brazilian cities and the Federal District show that the prevalence of adult obesity was 20.3%, with similar rates for men and women [6]. This statistic may indicate a shift in the pattern of weight gain in the Brazilian population with respect to sex, at least in the setting of major metropoles.

The obesity transition can be classified into four. Stage 1 is characterized by a higher prevalence of obesity in women with lower education at a lower economic level than those with a lower socioeconomic level. In stage 2 there is a large increase in the prevalence of obesity among adults, a smaller one among children, and an increase in the distance between the sexes and an increase in socioeconomic differences among women. At stage 3, a prevalence of obesity between those with lower socioeconomic status and those with higher socioeconomic status, and a platform in the prevalence, can be observed in women of high socioeconomic status and children. The putative final stage (stage 4) of the obesity transition would be a declining prevalence [22]. Considering the data analyzed in the present study, it can be considered that, in 2013, Brazil was in stage 1 of the obesity transition with a higher percentage of weight gain for females when compared to males.

Besides the difference in weight gain between sexes, differences across age groups were also evident. Among the Brazilian population studied, weight gain was not only found to be continuous, but the greater increase in body mass appeared to occur in the first decade of adulthood. Nonetheless, it is not possible to affirm that weight gain did not also occur at 30–34 years in older age groups, due to the cross-sectional character of the study. Moreover, the potential influence of the generational aspect between the age groups, a cohort effect, which might impact weight at 20 years, should also be taken into account. However, different national and international studies also pointed out that weight gain seems to be greater in the first decade of adult life [23,24,25,26].

Analyzing data from some major cities of Brazil, studies concluded that most individuals experience weight gain after age 20. In a sample of 875 individuals (30–59 years) from Goiânia city, the Goiás state capital, 63.6% exhibited weight gain of over 10% [23]. Another study, of 1341 adults (aged 21–30, 31–40, 41–50 and ≥51 years) from the city of Florianópolis, the Santa Catarina state capital, found that most men and women showed an increase of over 10% in BMI, while a quarter of women and 14% of men showed a BMI increase of over 30% [24]. In addition, a study based on seven cross-sectional surveys of Vigitel data, involving all major Brazilian cities, analyzed time trends for weight gain and change in nutritional status among men and women. In 2006, the greatest change in mean weight gain among males was observed in the 30–34 age bracket. By comparison, in 2012, the highest weight gain for men occurred in the 21–24 age bracket. For women, in 2006, the greatest increases in weight gain were found between 30–34 and 35–44 years (3.7 kg) and for the 35–44 and 45–59 (4.2 kg) age groups. In 2012, however, the highest increase in body weight occurred at 21–24 (3.4 kg) and between 21–24 and 25–29 years (5.1 kg) [25].

Similar results were reported for other populations. A North-American study of over 1800 individuals aged 20–39 years assessed the nutritional status by age group. The study found an obesity rate of 25.7% in the 20–24.9 years age group, 24.3% in the 25–29.9, 24% in the 30–34.9, and 26.0% in the 35–39.9 years age group [26]. Given that weight gain increases during the lifespan, the rate of obesity is expected to be higher among older individuals, a pattern also seen in Brazil. Data from the Vigitel system showed higher rates of obesity in older age groups [6]. In Australia, a cohort study found that mean change in BMI for both males and females was positively associated with the year of birth, where the prevalence of excess weight showed greater increases in cohorts born more recently than in older cohorts [27]. This cohort effect might be explained by dietary intake, given that younger individuals typically have a higher consumption of ultra-processed foods than older individuals [28].

The transition between adolescence and early adulthood is a key risk period for developing obesity in both sexes and different ethnic groups [18]. This might be explained by the fact the transition marks a period of major behavioral, social and economic complexity, in which different degrees of autonomy and responsibility manifest. Entry into the job market and leaving the parental home are examples of the responsibility acquired by these young adults [18,19]. At this stage in life, the young individual becomes responsible for their diet. The higher consumption of ultra-processed foods, which are quickly prepared and highly palatable, may explain the weight gain in this population [20], together with the lower level of physical activity, due to less time available for leisure. Engagement in physical activity has declined over the years in both sexes [21]. A cohort study involving North Americans found that weight gain during adulthood was associated with a significantly greater risk of developing chronic diseases (type 2 diabetes, hypertension and cancers associated to obesity) and a lesser probability of healthy aging [22].

The results in the present study failed to find any association between race/skin color and weight gain, which could be due to a lack of precision in the estimative, considering the small sample of some groups. However, in Brazil, the prevalence of obesity is known to be greater among black women than white women [29]. The higher prevalence of NCDs among black and mixed-race people indicates that risk factors must be distributed differently according to race/skin color and sex [30]. Although the black and mixed-race population, particularly female, is more likely to belong to the low socioeconomic strata, institutional racism is co-responsible for inequalities in the provision of care. This situation, among other factors, limits the action of professionals in the field of diagnosis and treatment, and also in the delivery of care for other health-related conditions [31,32,33]. Black women have a higher risk of death because racial and gender inequalities combine. These aspects are viewed as social and cultural constructions which define male and female roles, establishing hierarchies in which men are considered superior to women [30]. Furthermore, considering race/skin color as a social variable and not a biological one, the educational level could be better for adjusting weight gain in the PNS sample. Therefore, the relationship between race/skin color and nutritional status warrants further investigation in future studies.

In the present study, educational level was considered a proxy of income [34]. Women who had studied for 12 years or more showed a negative association with change in weight and BMI, i.e., women with a higher educational level gained less weight than women with fewer years of formal study, while the opposite held true for men. The association of nutritional status with education and income was found in a study exploring social inequity and malnutrition in all its forms (undernutrition and obesity) in the Brazilian population. Income and education exhibited similar associations to those found in the present study for men and women [29].

In Latin American countries, a collaborative work that described malnutrition in all its forms made it possible to understand the panorama of some middle-income countries. Overweight and obesity were the main problems of malnutrition in childhood and women. When observing the prevalence of overweight or obesity, as in our study, adult women with low schooling were 17 points more prevalent when compared to normal weight women. Therefore, schooling is a protective factor for adult women [35].

A review study performed in 2015 drawing on global data reported similar findings to those of the present study. A negative correlation between social class and family income with obesity was found. Women had consistently higher rates of overweight and obesity, the lower their income. The authors hypothesized that low-paid work for men generally involved greater physical effort and, therefore, men on lower incomes have higher energy expenditure than women in the same situation. Education also showed a strong negative association with overweight/obesity, especially in women [36].

Income played a transforming role regarding the risk of obesity last century. In the mid-twentieth century, wealth correlated with obesity in the USA and Europe, i.e., the greater one’s wealth, the higher the likelihood of having excess weight. In the last few decades, however, perhaps due to the abundance and wide availability of unhealthy foods in high-income countries, together with shifts in sociocultural norms, this relationship has changed. Currently, wealth in the USA tends to be negatively correlated with obesity, where individuals classified as poor, or below the poverty line, appear to have higher prevalence rates of obesity. Interestingly, in the present study, this pattern was not evident in males. Greater education showed a positive association with weight change in men. The factors underlying sex differences for this association remain unclear. Although the variables social class or income were not explored in the present study, a positive relationship exists between education and income, where individuals with more years of formal education are more likely to have a higher income [34]. The mechanisms by which income can impact health include: greater access to higher quality material resources, such as food and housing, better access to services that can improve health directly (such as health services and leisure activities), or indirectly, such as education and promotion of self-esteem, and socioeconomic position [37].

The variable area of residence showed a significant association only for men. Men living in rural areas tended to gain less weight than their urban counterparts. This pattern might be due to differing characteristics of foods, with greater consumption of ultra-processed foods in urban areas [38], or due to other aspects, such as frequency of occupational physical activity in the rural setting [39].

Evidence shows that, globally, there has been a steeper rise in BMI in rural areas than in urban areas. Between 1985 and 2017, the mean increase in BMI was 2.09 kg/m^2^ (95% CI 1.73; 2.44) and 2.10 kg/m^2^ (95% CI 1.79; 2.41) in women and men from rural areas, respectively, versus 1.35 kg/m^2^ (95% CI 1.05; 1.65) and 1.59 kg/m^2^ (95% CI 1.33; 1.84) in women and men living in urban settings [40]. In general, there has been a pattern of rising obesity rates in rural areas over recent years, a shift attributed to the modernization of society. Modernization has led to improvements in work tools and instruments, as well as to mechanization and automation in rural work, resulting in a reduction of work-related physical activity among rural dwellers [39]. However, the increase in weight, on average, is still greater in urban areas, suggesting that physical inactivity and/or poor diet are more common in this population. Despite the gap in the literature on the level of physical activity in individuals from rural and urban areas, studies show that the diet of rural dwellers is generally healthier than that of urban dwellers. According to a Brazilian study, individuals residing in rural areas are more likely to have a traditional diet that includes fresh or minimally-processed foods, particularly beans, together with a lower intake of ultra-processed foods, despite lower consumption of fruit and vegetables and fish [38].

The present study has some limitations. This was a cross-sectional study in which respondents assessed for events had different ages at the time of interview. Weight at 20 years of age was reported retrospectively and thus was susceptible to memory recall bias. Individuals from the older age groups may present a cumulative effect for this systematic error, leading to under or over-estimating of the weight reported. The number of indigenous and Asian individuals in the sample was low, hampering meaningful interpretation of differences relative to the other racial groups. Overall, however, the sample size and consistency and coherence with the literature support the validity of the findings of the present study.

The main strengths of the study are the national representativeness of the sample of 20,000 individuals, a wide diversity of the study population in terms of socioeconomic and demographic aspects, as well as the ability to stratify the respondents by sex. Lastly, the study was based on individual weight and BMI information for different time points, allowing the tracking of changes in nutritional status into adulthood. Therefore, the study results can add to the knowledge in the field by bridging the gap in the literature on weight gain in adulthood, an area which few Brazilian studies have explored to date [23,24,25,41].

## 5. Conclusions

The progressive weight gain seen over time contributed to the occurrence of obesity, representing a major public health issue, given the magnitude of the problem and negative health consequences. A higher weight gain was found among the younger group, low-educated women, and high-educated men from urban areas. These results point to the need to strengthen public policies that recognize the social determinants of health and that foster and promote the adoption of healthy lifestyles. Moreover, these results underscore the importance of caring for individuals with overweight and obesity, given that excess weight can lead to the development or exacerbation of other NCDs.

## Figures and Tables

**Table 1 ijerph-19-02851-t001:** Distribution of individuals for mean weight and body mass index (BMI), by sex. Brazil, 2013.

Sociodemographic Variables		Percentage in Sample % (95% CI)		Weight at 20 Years kg (95% CI)		Current Weight kg (95% CI)		BMI at 20 Years kg/m^2^ (95% CI)		Current BMI kg/m^2^ (95% CI)	
		Men	Women	Men	Women	Men	Women	Men	Women	Men	Women
Brazil		44.8 (44.1; 45.5)	55.1 (54.4; 55.8)	66.1 (65.7; 66.5)	53.9 (53.6; 54.2)	79.8 (79.2; 80.3)	69.6 (69.2; 70.1)	22.4 (22.3; 22.5)	21.3 (21.2; 21.4)	27.0 (26.9; 27.2)	27.5 (27.3; 27.7)
Age group	30–34 years	20.6 (19.3; 21.8)	21.3 (20.3; 22.4)	68.3 (67.3; 69.2)	55.4 (54.9; 56.0)	80.4 (79.3; 81.5)	67.6 (66.8; 68.5)	22.6 (22.3; 22.9)	21.4 (21.9; 21.7)	26.6 (26.3; 27.0)	26.2 (25.8; 26.5)
	35–39 years	17.7 (16.6; 18.9)	18.8 (17.8; 19.9)	66.2 (65.3; 67.1)	55.3 (54.5; 56.0)	79.8 (78.6; 81.1)	70.5 (69.3; 71.8)	22.2 (22.0; 22.5)	21.6 (21.3; 21.9)	26.8 (26.4; 27.1)	27.6 (27.1; 28.1)
	40–44 years	15.0 (13.9; 16.1)	16.8 (15.7; 17.9)	66.3 (65.5; 67.1)	54.0 (53.3; 54.6)	80.8 (79.5; 82.0)	69.4 (68.4; 70.4)	22.4 (22.2; 22.6)	21.3 (21.0; 21.5)	27.3 (26.9; 27.6)	27.3 (26.9; 27.7)
	45–50 years	16.6 (15.4; 17.9)	15.9 (14.8; 17.1)	65.3 (64.3; 66.3)	52.8 (52.1; 53.6)	79.4 (78.0; 80.8)	70.7 (69.7; 71.8)	22.3 (22.0; 22.6)	21.0 (20.7; 21.3)	27.1 (26.7; 27.5)	28.1 (27.7; 28.5)
	51–54 years	15.8 (14.5; 17.3)	14.4 (13.3; 15.5)	64.7 (63.7; 65.6)	52.9 (52.2; 53.7)	78.6 (77.2; 80.0)	69.9 (68.9; 71.0)	22.3 (22.0; 22.5)	21.3 (21.0; 21.6)	27.0 (26.6; 27.4)	28.1 (27.7; 28.5)
	55–59 years	14.0 (12.8; 15.4)	12.5 (11.5; 13.5)	64.9 (63.3; 66.4)	51.8 (51.0; 52.6)	79.5 (77.7; 81.3)	70.1 (68.9; 71.4)	22.6 (22.0; 23.1)	21.1 (20.8; 21.4)	27.6 (27.6; 28.1)	28.5 (28.1; 29.0)
Race/Skin color	White	51.1 (49.5; 52.8)	52.9 (51.4; 54.4)	67.3 (66.7; 67.9)	54.2 (53.8; 54.6)	81.7 (80.9; 82.4)	70.1 (69.4; 70.7)	22.5 (22.3; 22.6)	21.2 (21.1; 21.4)	27.3 (27.1; 27.5)	27.4 (27.2; 27.7)
	Black	8.8 (7.7; 9.9)	8.2 (7.4; 9.1)	65.8 (63.9; 67.7)	55.3 (54.3; 56.3)	78.9 (76.6; 81.2)	72.5 (70.6; 74.3)	22.6 (22.0; 23.2)	21.6 (21.2; 21.9)	27.0 (26.4; 27.7)	28.3 (27.5; 29.1)
	Mixed-race	38.9 (37.2; 40.5)	37.1 (35.7; 38.6)	64.6 (64.0; 65.1)	53.3 (52.8; 53.7)	77.6 (76.7; 78.5)	68.5 (67.9; 69.2)	22.3 (21.1; 22.4)	21.4 (21.1; 21.6)	26.7 (26.5; 27.0)	27.5 (27.2; 27.7)
	Asian	0.7 (0.5; 1.0)	1.1 (0.8; 1.5)	66.0 (62.9; 69.2)	52.5 (50.7; 54.3)	73.8 (70.9; 76.6)	64.6 (60.7; 68.5)	23.0 (21.7; 24.2)	21.4 (20.7; 22.0)	25.6 (24.6; 26.7)	26.2 (24.8; 27.7)
	Indigenous	0.3 (0.2; 0.5)	0.5 (0.3; 0.7)	61.8 (58.4; 65.1)	52.2 (49.8; 54.6)	76.4 (72.0; 80.9)	68.3 (62.5; 74.0)	21.9 (20.8; 23.0)	21.6 (20.5; 22.6)	27.1 (25.7; 28.5)	28.2 (25.9; 3.5)
Educational level	0–8 years	34.5 (32.6; 36.4)	28.7 (27.3; 30.3)	63.6 (62.9; 64.2)	53.3 (52.7; 53.9)	76.0 (75.0; 77.1)	70.0 (69.2; 70.8)	22.2 (21.9; 22.4)	21.7 (21.4; 21.9)	26.4 (26.1; 26.8)	28.4 (28.1; 28.8)
	9–11 years	14.8 (13.5; 16.3)	14.5 (13.5; 15.5)	66.5 (65.2; 67.8)	53.3 (52.6; 54.0)	79.2 (77.8; 80.6)	70.1 (69.1; 71.1)	22.7 (22.3; 23.1)	21.3 (21.0; 21.5)	27.0 (26.6; 27.5)	28.0 (27.6; 28.4)
	≥12 years	50.7 (48.7; 52.7)	56.8 (55.1; 58.4)	67.6 (67.1; 68.2)	54.4 (54.0; 54.8)	82.5 (81.8; 83.2)	69.3 (68.7; 69.9)	22.5 (22.3; 22.6)	21.1 (21.0; 21.3)	27.4 (27.2; 27.6)	26.9 (26.7; 27.2)
Area of residence	Urban	87.2 (86.1; 88.1)	90.5 (89.7; 91.2)	66.4 (66.0; 66.9)	54.0 (53.7; 54.3)	80.6 (80.0; 81.2)	69.7 (69.2; 70.2)	22.4 (22.3; 22.6)	21.3 (21.2; 21.4)	27.2 (27.0; 27.4)	27.5 (27.3; 27.7)
	Rural	12.7 (11.8; 13.8)	9.4 (8.7; 10.2)	63.8 (62.8; 64.7)	53.6 (52.7; 54.5)	74.1 (73.0; 75.1)	69.0 (67.8; 70.1)	22.5 (21.9; 22.5)	21.5 (21.2; 21.8)	25.8 (25.4; 26.1)	27.7 (27.2; 28.2)

**Table 2 ijerph-19-02851-t002:** Mean change in weight and BMI, by sex. Brazil, 2013.

Sociodemographic Variables		Difference in Weight kg (95% CI)		Difference in BMI kg/m^2^ (95% CI)	
		Men	Women	Men	Women
Brazil		13.7 (13.1; 14.2)	15.6 (15.2; 16.0)	4.6 (4.4; 4.7)	6.2 (6.0; 6.3)
Age group	30–34 years	12.1 (11.2; 13.0)	12.1 (11.4; 12.9)	4.0 (3.7; 4.3)	4.7 (4.4; 5.0)
	35–39 years	13.6 (12.5; 14.6)	15.2 (14.2; 16.2)	4.5 (4.1; 4.8)	5.9 (5.5; 6.3)
	40–44 years	14.4 (13.3; 15.6)	15.4 (14.4; 16.3)	4.8 (4.4; 5.2)	6.0 (5.6; 6.4)
	45–50 years	14.0 (12.8; 15.2)	17.8 (16.9; 18.8)	4.7 (4.3; 5.1)	7.0 (6.7; 7.4)
	51–54 years	13.9 (12.5; 15.2)	17.0 (15.8; 18.1)	4.7 (4.3; 5.1)	6.8 (6.3; 7.2)
	55–59 years	14.6 (12.6; 16.5)	18.3 (17.0; 19.5)	5.0 (4.3; 5.6)	7.4 (6.9; 7.9)
Race/skin color	White	14.3 (13.7; 15.0)	15.8 (15.2; 16.3)	4.7 (4.5; 5.0)	6.2 (5.9; 6.4)
	Black	13.0 (11.0; 15.1)	17.1 (15.5; 18.8)	4.4 (3.7; 5.1)	6.7 (6.0; 7.3)
	Mixed-race	13.0 (12.2; 13.9)	15.2 (14.6; 15.8)	4.4 (4.1; 4.7)	6.1 (5.8; 6.3)
	Asian	7.7 (3.1; 12.3)	12.0 (8.9; 15.2)	2.6 (1.0; 4.2)	4.8 (3.6; 6.1)
	Indigenous	14.6 (11.1; 18.2)	16.0 (10.3; 21.8)	5.1 (3.8; 6.3)	6.6 (4.3; 8.9)
Educational level	0–8 years	12.4 (11.4; 13.4)	16.7 (15.9; 17.4)	4.2 (3.9; 4.6)	6.7 (6.4; 7.0)
	9–11 years	12.7 (11.2; 14.1)	16.8 (15.7; 17.8)	4.3 (3.8; 4.8)	6.7 (6.2; 7.1)
	≥12 years	14.8 (14.1; 15.5)	14.8 (14.3; 15.4)	4.9 (4.7; 5.1)	5.7 (5.5; 5.9)
Area of residence	Urban	14.2 (13.6; 14.7)	15.7 (15.2; 16.1)	4.7 (4.5; 4.9)	6.2 (6.0; 6.3)
	Rural	10.3 (9.4; 11.1)	15.3 (14.2; 16.4)	3.5 (3.2; 3.8)	6.1 (5.6; 6.6)

**Table 3 ijerph-19-02851-t003:** Percentage change in nutritional status (at age 20 years versus current age) in men. Brazil, 2013.

Sociodemographic Variables		Normal Weight at Age 20 (≤24.9 kg/m^2^)			Overweight at Age 20 (25.0–29.9 kg/m^2^)			Obesity at Age 20 (≥30.0 kg/m^2^)		
		Current Normal Weight	Current Overweight	Current Obesity	Current Normal Weight	Current Overweight	Current Obesity	Current Normal Weight	Current Overweight	Current Obesity
		% (95% CI)	% (95% CI)	% (95% CI)	% (95% CI)	% (95% CI)	% (95% CI)	% (95% CI)	% (95% CI)	% (95% CI)
Brazil		37.6 (35.5; 39.6)	45.4 (43.3; 47.5)	16.9 (15.5; 18.4)	17.3 (14.3; 20.7)	38.2 (34.3; 42.4)	44.3 (40.2; 48.6)	10.1 (5.3; 18.2)	26.5 (17.1; 38.6)	63.3 (50.2; 74.6)
Age group	30–34 years	43.9 (39.4; 48.4)	44.5 (39.9; 49.2)	11.5 (9.2; 14.3)	12.2 (6.7; 21.2)	44.9 (35.6; 54.7)	42.7 (33.7; 52.3)	2.1 (0.6; 6.6)	27.0 (14.2; 45.3)	70.8 (52.5; 84.1)
	35–39 years	41.1 (36.7; 45.5)	44.0 (39.6; 48.4)	14.8 (12.0; 18.1)	13.3 (8.7; 19.6)	38.2 (29.3; 47.9)	48.4 (39.5; 57.4)	26.5 (8.3; 58.8)	16.7 (6.9; 35.3)	56.7 (30.9; 79.2)
	40–44 years	35.3 (31.2; 39.7)	44.8 (40.4; 49.3)	19.7 (16.3; 23.5)	14.6 (9.4; 22.1)	39.7 (31.4; 48.7)	45.5 (36.4; 54.9)	5.9 (1.3; 22.2)	52.0 (22.9; 79.8)	41.9 (17.3; 71.2)
	45–50 years	36.9 (32.2; 41.9)	44.1 (39.4; 48.9)	18.8 (15.2; 23.0)	20.6 (13.8; 29.6)	32.4 (23.6; 42.7)	46.8 (36.6; 57.4)	18.1 (6.9; 39.7)	12.8 (4.1; 33.3)	69.0 (45.0; 85.8)
	51–54 years	34.0 (28.8; 39.7)	47.9 (42.2; 53.6)	17.9 (14.2; 22.3)	22.5 (13.7; 34.6)	39.5 (29.1; 51.0)	37.9 (27.5; 49.4)	2.5 (0.5; 11.4)	68.7 (38.7; 88.4)	28.6 (10.3; 58.2)
	55–59 years	30.9 (26.1; 36.0)	47.9 (42.1; 53.8)	21.1 (16.6; 26.3)	23.9 (15.3; 35.2)	31.5 (20.7; 44.7)	44.5 (31.5; 58.3)	7.9 (1.5; 32.7)	12.1 (3.4; 35.1)	79.8 (51.0; 93.7)
Race/skin color	White	34.3 (31.5; 37.2)	47.6 (44.6; 50.7)	18.0 (15.8; 20.3)	12.6 (9.2; 17.1)	39.3 (33.4; 45.6)	47.9 (41.8; 54.1)	12.7 (5.4; 27.0)	27.6 (14.7; 45.8)	59.6 (42.3; 74.8)
	Black	42.5 (35.8; 49.5)	38.0 (31.7; 44.8)	19.3 (15.2; 24.3)	25.0 (15.6; 37.4)	22.8 (13.9; 35.0)	52.1 (36.7; 67.1)	0.4 (0.0; 4.8)	12.8 (2.2; 48.3)	86.7 (51.0; 97.6)
	Mixed-race	43.3 (27.5; 60.6)	50.8 (33.8; 67.5)	5.8 (1.9; 16.3)	37.8 (9.1; 78.6)	49.6 (15.2; 84.4)	12.4 (2.7; 41.8)	0	100.0	0
	Asian	40.6 (37.7; 43.6)	44.1 (41.3; 47.0)	15.1 (13.3; 17.2)	21.4 (16.2; 27.6)	39.6 (33.8; 45.8)	38.8 (33.1; 44.8)	11.0 (4.8; 23.3)	29.4 (16.9; 46.0)	59.5 (42.9; 74.1)
	Indigenous	42.6 (23.2; 64.6)	39.6 (20.8; 62.2)	17.6 (5.9; 41.9)	6.3 (1.8; 19.6)	37.8 (13.1; 71.0)	55.7 (25.7; 82.0)	0	100.0	0
Educational level	0–8 years	41.5 (38.1; 45.0)	41.9 (38.6; 45.3)	16.4 (13.5; 20.8)	25.7 (12.7; 32.7)	39.1 (32.3; 46.4)	35.0 (28.1; 42.6)	26.7 (12.4; 48.3)	28.7 (15.3; 47.1)	44.5 (25.3; 65.5)
	9–11 years	40.0 (38.1; 45.0)	43.0 (38.0; 48.1)	16.9 (13.5; 20.8)	19.8 (12.2; 30.4)	35.0 (24.9; 46.7)	45.1 (33.8; 56.9)	7.6 (1.5; 30.8)	35.7 (10.1; 73.2)	56.6 (20.2; 87.0)
	≥12 years	34.1 (31.5; 36.8)	48.6 (45.7; 51.4)	17.2 (15.3; 19.4)	11.7 (8.3; 16.1)	38.5 (33.2; 44.1)	49.6 (44.0; 55.3)	2.8 (1.3; 5.9)	20.9 (11.7; 34.5)	76.1 (62.7; 85.8)
Area of residence	Urban	35.8 (33.5; 38.1)	46.3 (44.0; 48.6)	17.8 (16.2; 19.5)	16.6 (13.4; 20.2)	37.5 (33.3; 41.9)	45.8 (41.3; 50.3)	9.0 (4.3; 17.8)	26.7 (16.7; 39.8)	64.2 (50.3; 76.1)
	Rural	49.1 (44.9; 53.3)	39.9 (35.9; 44.0)	10.9 (8.9; 13.3)	24.0 (16.7; 33.2)	45.2 (34.9; 55.9)	30.6 (22.3; 40.5)	21.4 (6.6; 51.3)	24.7 (7.9; 55.5)	53.8 (21.7; 82.9)

**Table 4 ijerph-19-02851-t004:** Percentage change in nutritional status (at age 20 versus current age) in women. Brazil, 2013.

Sociodemographic Variables		Normal Weight at Age 20 (≤24.9 kg/m^2^)			Overweight at Age 20 (25.0–29.9 kg/m^2^)			Obesity at Age 20 (≥30.0 kg/m^2^)		
		Current Normal Weight	Current Overweight	Current Obesity	Current Normal Weight	Current Overweight	Current Obesity	Current Normal Weight	Current Overweight	Current Obesity
		% (95% CI)	% (95% CI)	% (95% CI)	% (95% CI)	% (95% CI)	% (95% CI)	% (95% CI)	% (95% CI)	% (95% CI)
Brazil		38.5 (37.0; 40.1)	37.1 (35.7; 38.6)	24.2 (22.8; 25.6)	16.4 (13.1; 20.2)	24.6 (20.4; 29.2)	58.9 (54.0; 63.7)	10.9 (6.8; 16.9)	27.5 (20.6; 35.8)	61.4 (52.5; 69.7)
Age group	30–34 years	53.7 (50.4; 56.9)	32.5 (29.4; 35.8)	13.7 (11.4; 16.3)	15.4 (9.0; 25.2)	22.6 (15.0; 32.6)	61.8 (51.3; 71.4)	7.3 (2.8; 17.7)	38.9 (22.9; 57.7)	53.7 (36.1; 70.4)
	35–39 years	41.2 (37.8; 44.6)	36.5 (33.2; 39.8)	22.2 (19.3; 25.4)	12.8 (8.2; 19.3)	23.1 (15.6; 32.8)	63.9 (52.9; 73.6)	9.6 (3.1; 26.0)	23.1 (12.4; 39.0)	67.1 (49.8; 80.7)
	40–44 years	38.2 (34.6; 41.9)	36.7 (33.3; 40.2)	25.0 (21.8; 28.5)	22.3 (13.6; 34.4)	25.3 (17.6; 35.0)	52.2 (41.0; 63.3)	12.7 (4.0; 33.5)	24.5 (10.4; 47.4)	62.6 (40.2; 80.7)
	45–50 years	31.8 (28.0; 35.8)	41.2 (37.3; 45.2)	26.8 (23.2; 30.7)	13.8 (8.4; 21.8)	19.3 (10.9; 31.9)	66.7 (54.5; 77.1)	13.8 (4.2; 36.7)	22.5 (9.0; 46.0)	63.5 (37.3; 83.6)
	51–54 years	30.0 (26.2; 34.1)	41.2 (36.9; 45.6)	28.7 (24.9; 32.7)	20.3 (11.1; 34.1)	27.4 (15.3; 44.1)	52.2 (37.4; 66.6)	11.9 (3.7; 32.3)	12.8 (3.2; 39.3)	75.1 (49.0; 90.5)
	55–59 years	28.5 (24.6; 32.6)	36.6 (32.3; 41.0)	34.8 (30.2; 39.7)	15.1 (8.8; 24.8)	34.4 (22.3; 48.9)	50.3 (37.2; 63.4)	12.1 (2.9; 38.5)	40.0 (18.4; 66.4)	47.8 (22.3; 74.5)
Race/skin color	White	39.7 (37.6; 41.9)	36.2 (34.0; 38.4)	23.9 (21.9; 26.0)	13.6 (10.1; 18.1)	27.5 (21.5; 34.6)	58.7 (51.7; 65.4)	14.6 (7.6; 26.2)	30.1 (19.9; 42.8)	55.1 (41.7; 67.8)
	Black	34.5 (29.4; 39.9)	35.0 (30.2; 40.2)	30.3 (25.4; 35.7)	15.8 (6.4; 33.9)	16.7 (9.1; 28.7)	67.3 (50.3; 80.7)	17.2 (4.3; 48.8)	19.3 (6.3; 45.8)	63.3 (37.6; 83.1)
	Mixed-race	52.9 (37.3; 67.8)	33.4 (20.5; 49.3)	13.6 (6.0; 27.8)	31.0 (9.7; 65.1)	2.6 (0.3; 18.1)	66.3 (32.8; 88.7)	0	0	100.0
	Asian	37.1 (34.7; 39.6)	39.2 (36.8; 41.7)	23.5 (21.5; 25.6)	19.5 (14.1; 26.4)	23.8 (17.4; 31.6)	56.6 (48.6; 64.2)	6.5 (3.2; 12.9)	26.1 (16.0; 39.5)	67.2 (53.5; 78.5)
	Indigenous	45.2 (27.1; 64.6)	26.9 (15.2; 43.0)	27.8 (14.7; 46.4)	0.2 (0.0; 2.7)	1.7 (0.2; 10.1)	97.9 (89.6; 99.6)	0	98.6 (88.6; 99.8)	13.7 (0.1; 11.3)
Educational level	0–8 years	31.0 (28.4; 33.7)	39.7 (36.8; 42.8)	29.1 (26.5; 31.9)	19.2 (13.7; 26.2)	25.5 (18.4; 34.2)	55.1 (47.2; 62.8)	15.2 (7.9; 27.2)	22.2 (12.7; 35.8)	62.5 (47.5; 75.4)
	9–11 years	32.5 (28.8; 36.3)	39.1 (35.2; 43.1)	28.3 (24.8; 32.2)	14.3 (8.7; 22.2)	21.9 (14.3; 32.0)	63.7 (52.5; 73.5)	17.9 (7.6; 36.3)	22.9 (10.2; 43.7)	59.1 (37.5; 77.6)
	≥12 years	43.6 (41.5; 45.8)	35.4 (33.5; 37.4)	20.8 (19.1; 22.7)	14.6 (10.4; 20.2)	24.5 (19.3; 30.0)	60.7 (53.7; 67.2)	3.4 (1.4; 7.7)	35.5 (24.7; 47.9)	61.0 (48.6; 72.0)
Area of residence	Urban	38.9 (37.3; 40.6)	36.9 (35.3; 38.5)	24.1 (22.6; 25.6)	16.3 (12.8; 20.5)	22.8 (18.8; 27.4)	60.7 (55.6; 65.6)	11.3 (6.9; 18.0)	28.3 (20.8; 37.3)	60.2 (50.6; 69.1)
	Rural	34.9 (30.8; 39.2)	39.6 (35.7; 43.6)	25.4 (21.7; 29.4)	17.1 (10.2; 27.3)	38.8 (24.0; 56.0)	43.9 (30.2; 58.5)	6.7 (1.9; 20.4)	19.8 (7.3; 43.6)	73.4 (50.1; 88.3)

**Table 5 ijerph-19-02851-t005:** Association of sociodemographic variables with weight and BMI increase, by sex. Brazil, 2013.

Sociodemographic Variables		Weight Increase				BMI Increase			
		Crude Coefficient (95% CI)		Adjusted Coefficient (95% CI)		Crude Coefficient (95% CI)		Adjusted Coefficient (95% CI)	
		Men	Women	Men *	Women **	Men	Women	Men *	Women **
Age in full years		0.07 (0.01; 0.13)	0.22 (0.18; 0.27)	0.10 (0.04; 0.16)	0.22 (0.17; 0.26)	0.03 (0.01; 0.05)	0.10 (0.08; 0.12)	0.04 (0.02; 0.06)	0.09 (0.07; 0.11)
Race/skin color	White	Reference	Reference			Reference	Reference		
	Black	−1.30 (−3.46; 0.86)	1.35 (−0.4; 3.10)			−0.35 (−1.06; 0.35)	0.05 (−0.19; 1.20)		
	Mixed-race	−1.31 (−2.36; −0.26)	−0.57 (−1.40; 0.26)			−0.33 (−0.68; 0.01)	−0.09 (−0.42; 0.24)		
	Asian	−6.62 (−11.23; −2.02)	−3.73 (−6.93; −0.50)			−2.12 (−3.72; −0.52)	−1.33 (−2.61; −0.04)		
	Indigenous	0.28 (−3.32; 3.89)	0.24 (−5.53; 6.01)			0.34 (−0.93; 1.61)	0.44 (−1.91; 2.80)		
Educational level	0–8 years	Reference	Reference	Reference	Reference	Reference	Reference	Reference	Reference
	9–11 years	0.24 (−1.52; 2.02)	0.11 (−1.21; 1.44)	0.02 (−1.83; 1.79)	0.59 (−0.73; 1.93)	0.03 (−0.55; 0.63)	−0.03 (−0.57; 0.50)	−0.03 (−0.64; 0.56)	0.17 (−0.36; 0.71)
	≥12 years	2.39 (1.17; 3.61)	−1.80 (−2.73; −0.93)	2.12 (0.82; 3.41)	−1.04 (−1.95; −1.38)	0.64 (−0.23; 1.05)	−0.98 (−1.33; −0.62)	0.58 (0.15; 1.01)	−0.63 (−0.99; −0.27)
Area of residence	Urban	Reference	Reference	Reference		Reference	Reference	Reference	
	Rural	−3.89 (−4.94; −2.85)	−0.38 (−1.60; 0.83)	−3.13 (−4.3; −1.94)		−1.21 (−1.57; −0.85)	−0.04 (−0.55; 0.45)	−1.02 (−1.42; −0.61)	

* Adjusted for age, education, and area of residence. ** Adjusted for age and education.

## Data Availability

The data that support the findings of this study are openly available in the Brazilian Institute of Geography and Statistics (IBGE) at https://www.ibge.gov.br/estatisticas/sociais/saude/9160-pesquisa-nacional-de-saude.html?=&t=microdados (accessed on 20 February 2022).

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
