# Peer review of "Weight Gain and Change in Body Mass Index after Age 20 in the Brazilian Population and Associated Sociodemographic Factors: Data from the National Health Survey"

_ijerph, 2022, doi:10.3390/ijerph19052851_

Round 1

Reviewer 1 Report

Thank you for the opportunity to review “Weight gain and change in body mass index after age 20 in the Brazilian population and associated sociodemographic factors: Data from the National Health Survey”. The study addresses an important topic and utilizes an expansive and representative sample. However, based on limitations of the data collection, some of the conclusions go beyond what is shown in the data and the presentation of the data lacks focus. I have included some suggestions below to improve the manuscript.

Major Issues:

  • The abstract and the introduction are set up to suggest that this manuscript will be about young adult weight gain; however, this is a finding from the analysis and does not seem to be the primary research question. I would recommend saving this for the discussion.
  • Conclusions regarding weight gain being greatest in young adulthood seem to overstep the data, which is cross-sectional. For instance, it cannot be concluded that the rate at which people ages 30-34 are gaining weight will slow down nor could one conclude that people aged 55-59 gained any weight between 20 and 30 years old. This is noted as a limitation, but it is a significant limitation, particularly given the conclusions that the authors make.
  • The statistics are somewhat unclear. What were the statistical tests used to compare the groups? Were there adjustments made for the multiple comparisons among groups?
  • There is a lot of data presented in this manuscript and it becomes difficult to follow at times. The paper would benefit from clearly specified aims and subheadings for the results that corresponded with the aims. Relatedly, if the weight and BMI data tell the same story, it may be more efficient to just discuss one metric in the text and indicate that the other showed similar results.
  • Early in the discussion, the authors talk about the differences between men and women in the sample; however, these groups were never compared in the analysis. Some of this discussion could potentially be used to justify the stratification in the methods of the paper, but doesn’t belong in the discussion.

Minor Issues:

  • Please indicate how urban or rural were defined
  • The term “yellow” to describe race/skin color is not a familiar term. Is there another way to describe this group?
  • If assessing weight gain in young adulthood was of interest, it is not clear to me why respondents between ages 20 and 30 were not included.
  • Please include some indication of significant relationships in the tables
  • Lines 125-128 need more information on what was included in the adjusted models and why
  • Line 269 – the conceptual obesity transition model is mentioned but not put into context. Additional detail would be beneficial if this is retained
  • Line 296-297 – Looking at the numbers presented, it’s not clear that this supports early adulthood as a time for more rapid weight gain (if that is indeed what the authors are hoping to show)
  • Lines 308 – 321 – there were no differences by race as the authors note. This discussion would be more helpful if it centered around WHY no differences were found in the current study. As currently written, it mostly lists reasons why one would expect to find them.
  • Revise for small grammar mistakes throughout

Reviewer 2 Report

Lines 31- 32. This is a powerful statement and needs proper justification.

Line 36. The citation format of cites 2 and 3 is not correct.

Line 52. Although not a sociodemographic factor I think that genetics has to be at least named as a risk factor.

Line 103. The authors did not report BMI under 18.5?

Line 203. I would try to avoid using the term “healthy weight” considering that no all individuals with <25 BMI are healthy (for example <18.5 BMI, clearly not healthy).

Table 5. Weight increase in men, why did the authors exclude race from the adjustment? Crude coefficients for mixed and yellow are both statistically significant. The same question for women, crude coefficient for yellow is different from 0.

Also Table 5. BMI increase. The yellow race/skin color present a significant coefficient in both sexes but was excluded from the adjusted model. Why? Also in men, the Education level showed no significant coefficients but was included in the adjuted model.

There is no mention of any nutritional study from a regional country (such as Argentina, Uruguay, Paraguay, etc.). It would be interesting if the author could at least mention the nutritional state of adults in a nearby country.
